# A Study on Verification of Waterproofing Method Properties for Performance Grading in Apartment Houses-Upper Slab of the Underground Structure

Ki-won An [1] and Byoung-il Kim [2],*

1   Institute of Construction Technology, Seoul National University of Science & Technology, 232 Gongneung-ro, Nowon-gu, Seoul 01811, Republic of Korea; ankiwon@seoultech.ac.kr
2   School of Architecture, Seoul National University of Science & Technology, 232 Gongneung-ro, Nowon-gu, Seoul 01811, Republic of Korea
*   Correspondence: bikim@seoultech.ac.kr; Tel.: +82-2-970-6512

**Abstract:** This study evaluates waterproofing methods for underground structures in multi-unit residential buildings. The objective is to select effective methods and establish performance ratings. Experimental assessments are conducted and scores are assigned for ranking. In this regard, waterproofing methods used in the upper slab of the top floor of multi-unit residential building underground structures were investigated, and they were categorized as composite waterproofing, sheet waterproofing, and membrane waterproofing methods. For performance evaluation purposes, experiments were conducted on qualified materials and scores were assigned to each test specimen, with a total score of 100 points. Based on the test results, scores for disqualified materials were deducted and, ultimately, the rankings of waterproofing materials and methods for the upper slab of the top floor of underground structures were determined based on the total aggregated scores. Subsequently, through proximity performance evaluation, the applied waterproofing methods were experimentally verified for defect issues based on the presence or absence of a proximity layer. This led to the ranking and performance grading of a total of 12 materials. The results confirmed the necessity of a proximity layer-dependent waterproofing method and highlighted the superiority of the composite waterproofing method with a proximity layer. Additionally, differences in installation methods and material properties between sheet and membrane waterproofing methods were identified, resulting in variations in performance grading.

**Keywords:** performance grade; waterproofing; basement top slab; root penetration resistance; evaluation technique

## 1. Introduction

In places where projects such as new city development and urban improvement plans are carried out in Korea, apartment construction plans are generally undertaken to accommodate a large number of people, and apartment buildings often become subjects of lawsuits due to various disputes and complaints [1]. To address this issue, the Ministry of Land, Infrastructure and Transport implemented the "Performance Grade Indication System for Apartment Houses." This system allows residents to check and select apartment houses based on their performance, and it notifies residents of the performance ratings [2,3]. The system evaluates five aspects: "noise performance", "structural performance", "environmental performance", "living environment performance", and "fire and fire-fighting performance", providing performance information in advance [4].

Despite the operation of the above system, the number of disputes and lawsuits related to leakage defects in apartment buildings has been increasing every year. Especially in large underground parking lots, which are becoming more important due to their size and space [5–7], the lack of guidelines and post-treatment measures for leakage defects

exacerbate the issue of underground space leakage [8,9]. The upper part of the top floor of underground spaces in apartment buildings, in particular, experiences high vibration or movement due to vehicle passage, making it susceptible to rainwater intrusion and leaks caused by cracks [10,11]. Additionally, the top floor of underground parking lots is constantly damaged not only by leaks due to defects in the waterproof layer of the upper slab, but also by issues like soil or gravel weight causing leakage [12,13] (See Figure 1). Therefore, there is an urgent need for laws, institutional leak prevention devices, and guidelines for underground spaces in apartments in Korea [14,15].

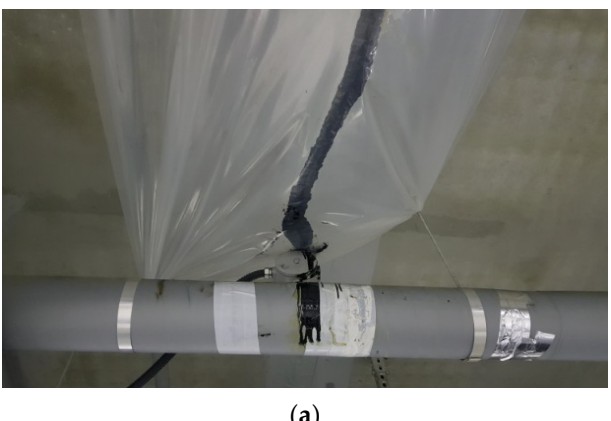 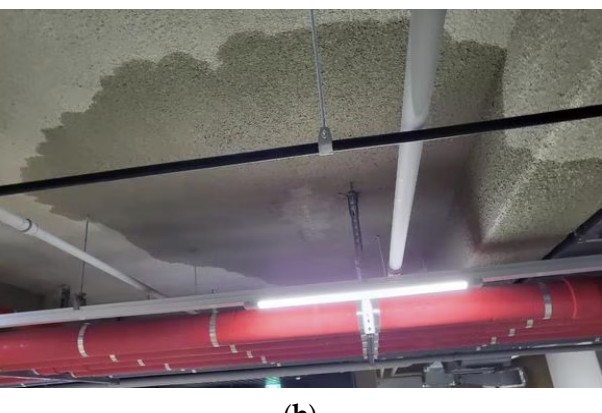

(**a**)        (**b**)

**Figure 1.** Defects in the top floor of an apartment building's basement structure. They should be listed as: (**a**) upper slab oil leakage; (**b**) upper slab leakage.

The objective of this study is to select waterproofing methods for the uppermost slab of underground structures in multi-unit residential buildings. This slab is directly exposed to external forces and prone to leakage damage. Experimental evaluations will be conducted to assess the effectiveness of the chosen waterproofing methods and performance rankings will be determined based on assigned scores. Additionally, the performance of each waterproofing method will be classified based on the presence or absence of a drainage layer. The findings will provide fundamental data for establishing waterproofing performance grades, aiming to prevent leakage incidents in multi-unit residential buildings.

## 2. Study on the Performance Rating Method of Apartment Housing

Korea's apartment performance rating system was first implemented in January 2006 and revised to enhance standards in July 2007 and January 2009. It was later removed as a housing performance rating requirement in February 2013 and has been operated as a green building certification system since June of the same year. The Ministry of Land, Infrastructure and Transport (Clause No. 2014-705) and the Ministry of Environment (Clause No. 2014-213) currently operate the system in a complex manner according to the "[Table 1] Apartment Certification Review Standards" [16].

The performance rating system for multi-unit residential buildings consists of 56 criteria, categorized into 26 essential items and 30 optional items. When establishing performance ratings, it was observed that 19 criteria are related to determining ratings based on material or method selection. Additionally, 14 criteria involve calculating performance ratios, while 9 criteria relate to upward adjustments of performance through the addition of lower-rated performances. Other criteria include rating based on area, distance, and material/construction quality, as well as performance measurement and scoring. Among these, the top three criteria account for 42 out of the total 56, representing approximately 75% of the criteria. Therefore, adopting these rating methods can contribute to stable utilization of performance ratings in the waterproofing field [17–23].

In conclusion, the performance ratings incorporated in the preliminary certification of environmentally friendly buildings for multi-unit residential developments are deter-

mined based on design drawings, plans, quality certification documents, and the use of high-performance measures [24,25]. Various methods such as the number of applicable materials, ratio per area, distance, scores, and upward adjustments of performance through the addition of materials at lower levels are utilized during the establishment of the rating system [26]. However, it was observed that the evaluation of performance ratings considers factors such as structure and environment but overlooks issues like environmental pollution resulting from the lack of waterproofing application and inconveniences faced by residents. Hence, establishing waterproofing performance ratings is necessary to address leakage prevention.

## 3. Verification Plan and Test Method for Performance Rating Method

*3.1. Checking the Waterproofing Method and Type of Test Evaluation*

In Korea, there is a method of constructing a waterproof layer on top of the slab (outer waterproofing method), and a composite waterproofing method and a single waterproofing method (sheets, coating film) are applied depending on the material applied. In addition, an additional root layer is constructed to prevent root intrusion of planting to protect the upper slab on the top floor of the underground structure. Accordingly, materials and methods applied to the upper slab of the top layer of the underground structure were classified into six materials (two composite waterproof materials, three sheet waterproof materials, and one coating waterproof material). The derived results are shown in Table 1 below.

**Table 1.** Types of Waterproofing Materials and Methods Used in the Upper Slab of Underground Structures.

| Item (Code) | Materials (Composition) | Note | Item (Code) | Materials (Composition) | Note |
|---|---|---|---|---|---|
| Material A | Composite Waterproofing Material (Adhesive Flexible Sealant + Modified Asphalt Sheet Waterproofing) + Root Penetration Resistance Layer | Composite Waterproofing Installation of Root Penetration Resistance Layer | Material G | Composite Waterproofing Material (Adhesive Flexible Sealant + Modified Asphalt Sheet Waterproofing) | Composite Waterproofing No Root Penetration Resistance Layer Installation |
| Material B | Composite Waterproofing Method (Polyurethane Membrane Waterproofing + Modified Asphalt Sheet Waterproofing) + Root Penetration Resistance Layer | Composite Waterproofing Installation of Root Penetration Resistance Layer | Material H | Composite Waterproofing Method (Polyurethane Membrane Waterproofing + Modified Asphalt Sheet Waterproofing) | Composite Waterproofing No Root Penetration Resistance Layer Installation |
| Material C | Self-adhesive Waterproofing Sheet (Rubber Asphalt-based or Butyl Rubber-based) + Root Penetration Resistance Layer | Sheet Waterproofing Installation of Root Penetration Resistance Layer | Material I | Self-adhesive Waterproofing Sheet (Rubber Asphalt-based or Butyl Rubber-based) | Sheet Waterproofing No Root Penetration Resistance Layer Installation |
| Material D | Modified Asphalt Waterproofing Sheet (Rubber Asphalt-based) + Root Penetration Resistance Layer | Sheet Waterproofing Installation of Root Penetration Resistance Layer | Material J | Modified Asphalt Waterproofing Sheet (Rubber Asphalt-based) | Sheet Waterproofing No Root Penetration Resistance Layer Installation |

**Table 1.** *Cont.*

| Item (Code) | Materials (Composition) | Note | Item (Code) | Materials (Composition) | Note |
|---|---|---|---|---|---|
| Material E | Synthetic Polymer Waterproofing Sheet (Vinyl Chloride Resin-based) + Root Penetration Resistance Layer | Sheet Waterproofing Installation of Root Penetration Resistance Layer | Material K | Synthetic Polymer Waterproofing Sheet (Vinyl Chloride Resin-based) | Sheet Waterproofing No Root Penetration Resistance Layer Installation |
| Material F | Membrane Waterproofing Material (Rubber Asphalt-based or Polyurethane Rubber-based) + Root Penetration Resistance Layer | Membrane Waterproofing Installation of Root Penetration Resistance Layer | Material L | Membrane Waterproofing Material (Rubber Asphalt-based or Polyurethane Rubber-based) | Membrane Waterproofing No Root Penetration Resistance Layer Installation |

As a result of examining the test and evaluation method for waterproof materials and construction methods for grading waterproof performance on the top floor slab of underground structures in apartments selected above, it was confirmed that there is a performance evaluation method for waterproof materials and construction methods operated by the Korean Construction Standards Center (KCSC). In addition, in order to check the performance according to the presence or absence of a root layer, the performance rating was classified in detail by applying the Korean Industrial Standard KS F 4938-18 "Rubber Performance Test Method". Accordingly, the test verification plan was established as shown in Table 2, utilizing the test evaluation approach of the Korean industrial standard KS F 4938-18 "Rubber Performance Test Method" and the test evaluation methodology outlined in KCS 11 44 00: 2018 Joint District.

**Table 2.** Verification Plan for Performance Grading of Upper Slab Materials in Underground Structures.

| Main Category | Middle Category | Subcategory | Verification Plan |
|---|---|---|---|
| Waterproofing Material | Composite Waterproofing Material | - | KCS 11 44 00: 2018 Evaluation of Performance for Waterproofing Materials and Methods in Joint District Structures |
| | Single Waterproofing Material | Sheet | |
| | | Membrane | |
| Root Penetration Resistance Performance | Root Penetration Resistance Layer | Installation | Korean Industrial Standard KS F 4938-18 "Test Methods for Root Penetration Resistance Performance" |
| | | Non-installation | |

*3.2. Test Method*

3.2.1. Performance Evaluation of Joint Residential Complex According to KCS 11 44 00: 2018

Waterproof materials must be applied to the upper slab on the lower floor of the underground structure of apartment houses, and methods such as sheets, coatings, and complex waterproofing are applied. In this situation, it is impossible to evaluate different materials according to specific criteria due to the large variation in physical properties for each material. Therefore, physical properties were verified to grade waterproof performance by applying the performance evaluation method of KCS 11 44 00: 2018 Joint District and tests were conducted on materials G to L without a root layer. The test methods conducted are shown in Table 3 below.

**Table 3.** Test Methods for Performance Verification of Upper Slab in Underground Structures.

| | Item | | Contents |
|---|---|---|---|
| 1 | Chemical Resistance Test | Test Specimen | • Mortar test specimen (Ø100 × 30 mm). <br> • Adhesion strength test specimen (210 × 70 × 30 mm). <br> • Application of waterproofing method on each test specimen. |
| | | Pre-treatment | • Immersion in water for 21 days. |
| | | Test Methods | • Verification of performance degradation such as blistering and delamination of the waterproofing layer. |
| 2 | Structural Response and Adaptability | Test Specimen | • Upper test specimen (Ø180 × 130 mm) including a fixing rod for a T-shaped test specimen with a height of 15 mm. <br> • Lower test specimen (Ø180 × 130 mm) with a cylindrical pipe (Ø40 × 130 mm) fixed in place. <br> • Application of waterproofing material between the upper and lower test specimens. |
| | | Pre-treatment | • Dried test specimen in fully dried condition after the completion of curing process. <br> • Moist test specimen in a moist condition after being submerged in water before the application of waterproofing material. |
| | | Test Methods | • Perform 100 cycles of movement at a speed of 50 mm/min with a displacement interval of 10.0 mm (±0.2 mm) by filling water at 20 °C temperature. <br> • After draining the water, subject the test specimen to 1 h of conditioning at −10 °C temperature. Then, perform 100 cycles of movement at a speed of 50 mm/min with a displacement interval of 10.0 mm (±0.2 mm). <br> • Fill water again at 20 °C temperature and perform 100 cycles of movement at a speed of 50 mm/min with a displacement interval of 10.0 mm (±0.2 mm). |
| 3 | Moisture Adhesion Performance Test | Test Specimen | • Dry test specimen of 300 × 300 × 50 mm concrete block. <br> • Moist test specimen of 300 × 300 × 50 mm concrete block. <br> • Application of waterproofing method on each test specimen. |
| | | Pre-treatment | • Untreated: test conducted without any pre-treatment. <br> • Freeze–thaw treatment: according to Korean Industrial Standard KS F 2456-18 "Test Method for Resistance of Concrete to Rapid Freezing and Thawing," perform 200 cycles of rapid freezing and thawing using Method B. <br> • Drying–wetting treatment: condition the specimen at 60 ± 2 °C for 1 h, followed by immersion in water at 20 ± 2 °C for 3 h, constituting 1 cycle. Repeat this process for 50 cycles. <br> • Thermal cycling treatment: increase the temperature from 20 °C to 60 °C for 1 h and 30 min, hold at 60 °C for 2 h, then decrease the temperature from 60 °C to −20 °C for 1 h and 30 min, hold at −20 °C for 2 h. Increase the temperature from −20 °C to 20 °C for 1 h and 30 min, hold at 20 °C for 2 h. This constitutes 1 cycle. Repeat this process for 50 cycles. <br> • Long-term immersion: submerge the specimen in water at 20 ± 3 °C for 30 days, then dry at room temperature for 24 h. |
| | | Test Methods | • Check for significant changes such as blistering, delamination, discoloration, and absorption in the central and edge areas of the test specimens. |

<p style="text-align:center"><b>Table 3.</b> <i>Cont.</i></p>

| Item | | Contents |
|---|---|---|
| 4 Water Tightness | Test Specimen | • Apply the waterproofing method on the mortar test specimen (Ø100 × 30 mm). |
| | Pre-treatment | • Conduct drying-saltwater immersion treatment: condition the specimen at 60 ± 2 °C for 1 h, followed by immersion in on-site saline water (20 ± 2 °C) for 3 h. Repeat this process for 50 cycles.<br>• Conduct long-term saline water immersion: submerge the specimen in on-site saline water (20 ± 2 °C) for 30 days, then dry at room temperature for 24 h. |
| | Test Methods | • Perform water penetration test according to Korean Industrial Standard KS F 4919 "Test Method for Cementitious Polymer Waterproofing Material" at 0.3 N/mm$^2$ pressure for 3 h. |
| 5 Temperature Dependency | Test Specimen | • Apply the waterproofing method on the mortar test specimen (Ø100 × 30 mm). |
| | Pre-treatment | • Untreated: conduct the test without any pre-treatment.<br>• Freeze–thaw treatment: follow Method B of Korean Industrial Standard KS F 2456-18 "Test Method for Resistance of Concrete to Rapid Freezing and Thawing" and perform 200 cycles of rapid freezing and thawing.<br>• Drying–wetting treatment: condition the specimen at 60 ± 2 °C for 1 h, followed by immersion in water at 20 ± 2 °C for 3 h, constituting 1 cycle. Repeat this process for 50 cycles.<br>• Thermal cycling treatment: increase the temperature from 20 °C to 60 °C for 1 h and 30 min, hold at 60 °C for 2 h, then decrease the temperature from 60 °C to −20 °C for 1 h and 30 min, hold at −20 °C for 2 h. Increase the temperature from −20 °C to 20 °C for 1 h and 30 min, hold at 20 °C for 2 h. This constitutes 1 cycle. Repeat this process for 50 cycles.<br>• Long-term immersion: submerge the specimen in water at 20 ± 3 °C for 30 days, then dry at room temperature for 24 h. |
| | Test Methods | • Perform water penetration test according to Korean Industrial Standard KS F 4919 "Test Method for Cementitious Polymer Waterproofing Material" at 0.3 N/mm$^2$ pressure for 3 h. |
| 6 Crack Resistance | Test Specimen | • Apply the waterproofing method on the mortar test specimen (40 × 40 × 160 mm). |
| | Pre-treatment | • Condition the specimen at −20 °C, 20 °C, and 60 °C for a minimum of 2 h each. |
| | Test Methods | • Conduct flexural strength test at a distance of 100 mm between support points, with a loading rate of 1 mm/min.<br>• Observe the occurrence of material failure, cracking, or any other signs of damage when the substrate fractures. |
| 7 Durability | Test Methods | • Verify the performance levels specified in the Korean Industrial Standard (KS) specifications or equivalent national standards for the main materials comprising the waterproofing layer. |
| 8 Low-temperature Adhesion Stability | Pre-treatment | • Use a Ø100 × 30 mm circular mortar substrate with a 3 mm hole drilled.<br>• Condition the test specimen in chambers set at 20 °C, 0 °C, −5 °C, and −10 °C for 1 h before applying the waterproofing material. |
| | Test Methods | • Perform water penetration test according to Korean Industrial Standard KS F 4919 "Test Method for Cementitious Polymer Waterproofing Material" at 0.3 N/mm$^2$ pressure for 3 h and observe any changes in the test specimen. |

### 3.2.2. Korea Industrial Standard KS F 4938-"18" "Standard Test Method For Root Penetration Resistance"

The root-proof performance test method is evaluated by applying each waterproof method and root-proof method inside a perforated root test body that can observe 800 × 800 × 220 mm of planting, planting it, and growing the plant for 24 months so that the root of the plant can grow sufficiently(Refer to Figure 2).

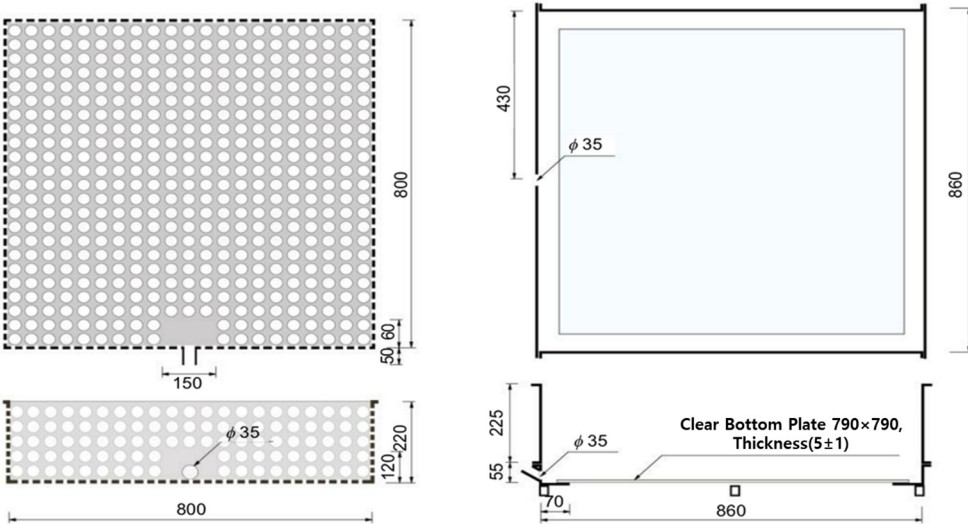

**Figure 2.** KS F 4938- "18" "Standard test method for root penetration resistance" Test equipment.

## 4. Results of Waterproof Performance Rating for Underground Structures in Apartment Buildings

*4.1. KCS 11 44 00: 2018 "Joint District" Performance Evaluation Test Results*

4.1.1. KCS 11 44 00: 2018 "Joint District" Evaluation Index Checked

KCS 11 44 00: 2018 "Joint District" was allocated to diversify the scores by deducting points from the number of failed tests so that quantitative performance results could be derived from the original evaluation index and the scores were set as points according to the evaluation results from 1 to 8 in Table 4 below.

**Table 4.** KCS 11 44 00: 2018 "Joint District" Evaluation Criteria and Scoring.

| Sequence | Item | Number of Test Specimens | Evaluation Criteria | | Score | |
| --- | --- | --- | --- | --- | --- | --- |
| | | | Initial Evaluation Criteria | This Study | Maximum Score | Minimum Score |
| 1 | Chemical Resistance Test | 6 | No performance degradation | Deduction of 3 points per test specimen for performance degradation. | 20 Point | 2 Point |
| | | | Performance degradation observed | | | |
| 2 | Structural Response and Adaptability | 6 | No leakage in all specimens | Deduction of 3 points per test specimen for leakage. | 20 Point | 2 Point |
| | | | 1 specimen leaking | | | |
| | | | 2 specimens leaking | | | |
| | | | 3 specimens leaking | | | |
| | | | 4 specimens leaking | | | |

**Table 4.** *Cont.*

| Sequence | Item | Number of Test Specimens | Evaluation Criteria | | Score | |
| --- | --- | --- | --- | --- | --- | --- |
| | | | Initial Evaluation Criteria | This Study | Maximum Score | Minimum Score |
| 3 | Moisture Adhesion Performance Test | 6 | Overall stability across the entire surface | Deduction of 1 point per test specimen for displacement. | 10 Point | 4 Point |
| | | | Edge delamination | | | |
| | | | Delamination at both the edges and center | | | |
| 4 | Water Tightness | 6 | No penetration | Deduction of 1 point per test specimen for permeability. | 10 Point | 4 Point |
| | | | Penetration occurred | | | |
| 5 | Temperature Dependency | 18 | Low deformation rate | Deduction of 0.5 points per test specimen for microcracks, discoloration, displacement, cracking, delamination, and permeation occurrences. | 10 Point | 1 Point |
| | | | Moderate deformation | | | |
| | | | High deformation rate | | | |
| 6 | Crack Resistance | 9 | Top 20% performance | Deduction of 1 point per test specimen for waterproofing material detachment. | 10 Point | 1 Point |
| | | | Middle 60% performance | | | |
| | | | Bottom 20% performance | | | |
| 7 | Durability | - | Top 20% performance | Deduction of 2 points per test specimen for failure to meet material-specific quality standards. | 10 Point | 0 Point |
| | | | Middle 60% performance | | | |
| | | | Bottom 20% performance | | | |
| 8 | Low-Temperature Adhesion Stability | 12 | Overall stability across the entire surface | Deduction of 0.5 points per test specimen for displacement. | 10 Point | 4 Point |
| | | | Edge delamination | | | |
| | | | Delamination at both the edges and center | | | |
| | Total Score | | | | 100 Point | 16 Point |

### 4.1.2. Chemical Settlement Stability

The test results for chemical immersion stability are shown in Figures 3 and 4 below. As a result of the chemical immersion stability test, material G was identified as having 17 points due to the excitation between materials in one test piece, and material H was identified as having 14 points due to the excitation and swelling of joints in two tests. In addition, material I was identified as having eight points due to joint excitation and end excitation in two test pieces, and material J was identified as having eight points due to joint excitation in four test pieces. Material K was identified as having 11 points due to bond excitation in three test pieces and, finally, swelling occurred in two test pieces of material L, giving it 14 points.

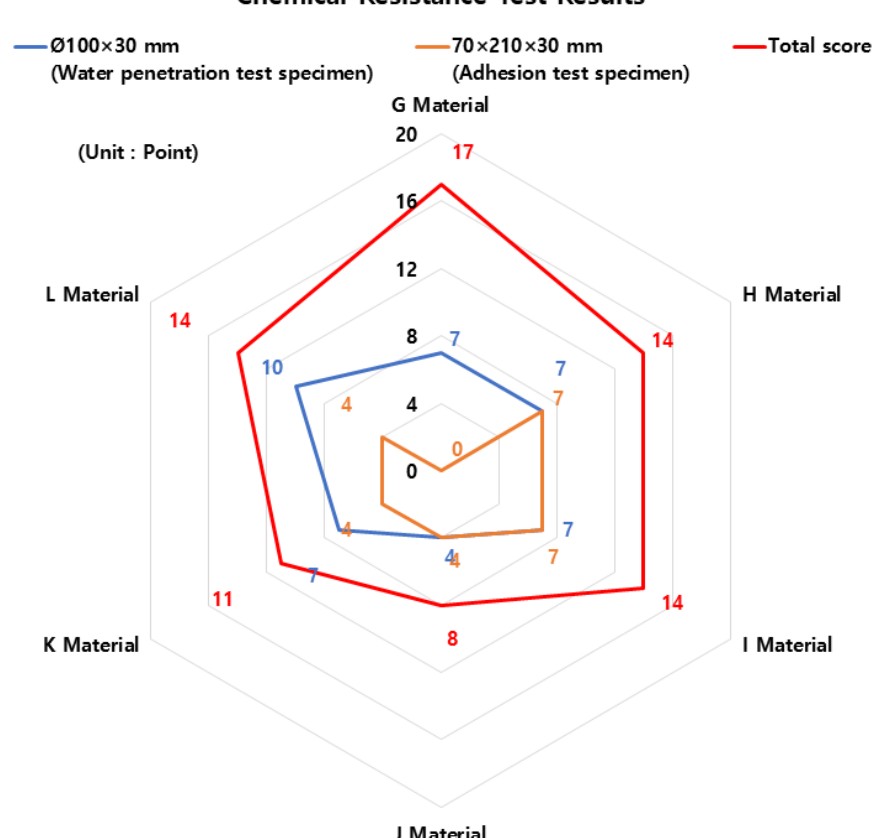

**Figure 3.** KCS 11 44 00: 2018 "Joint District" Chemical Resistance Test Results.

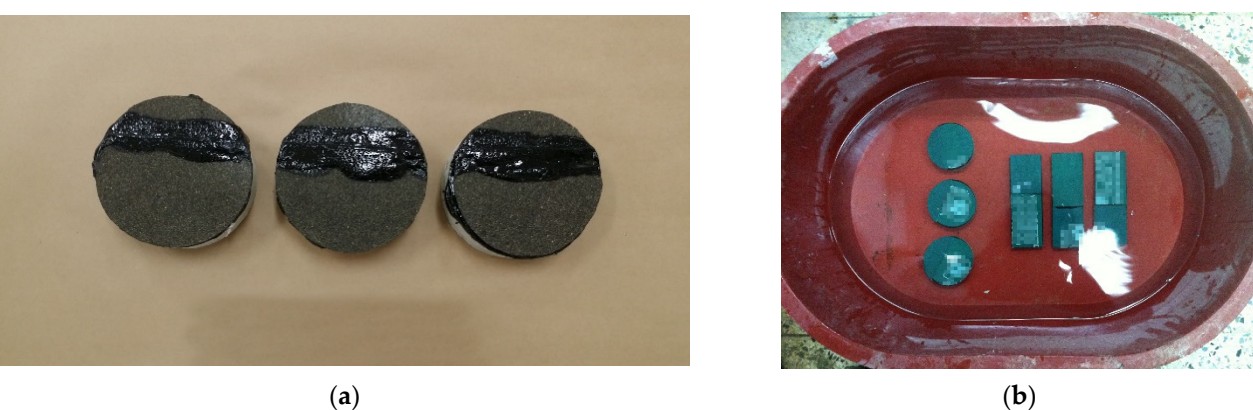

(**a**) (**b**)

**Figure 4.** KCS 11 44 00: 2018 "Joint District" Chemical Resistance Test Results. They should be listed as: (**a**) Chemical Resistance Test Specimens; (**b**) Chemical Resistance Test Status.

4.1.3. Structural Response and Adaptability

Figures 5 and 6 show the test results for Structural Response and Adaptability. As a result of the Structural Response and Adaptability test, material G leaked from one wet surface, and material H leaked from one wet surface to 17 points for both the lowest floor slab and the outer wall. In addition, leakage occurred in two wet surfaces and one dry surface of material I, and both the lowest floor slab and the outer wall were identified as having 11 points, and material J, material K, and material L were identified as having 2 points.

## Structural Response and Adaptability Test Results

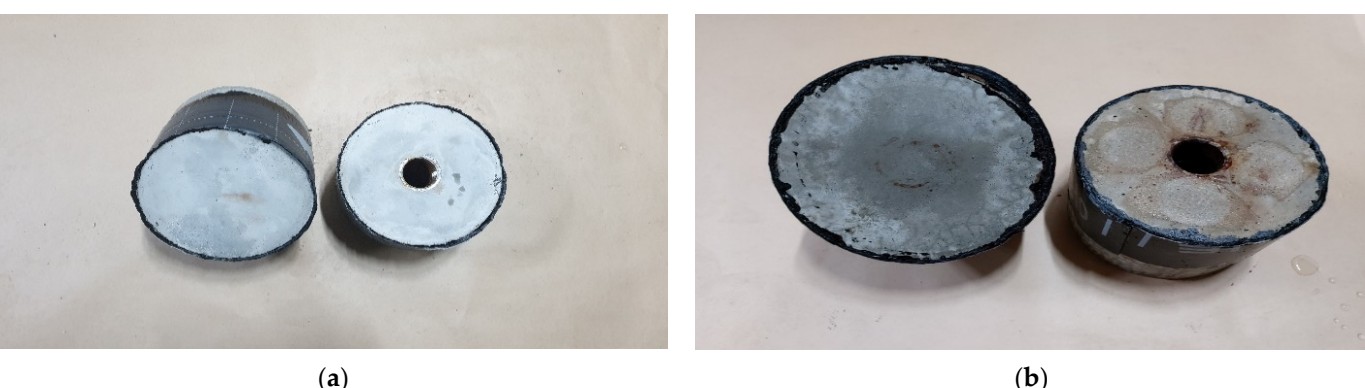

**Figure 5.** KCS 11 44 00: 2018 "Joint District" Chemical Resistance Test Results.

| (**a**) | (**b**) |
|---|---|

**Figure 6.** KCS 11 44 00: 2018 "Joint District" Structural Response and Adaptability Test Results. They should be listed as: (**a**) non-leakage test specimen (surface dry); (**b**) leakage test specimen (surface moist state).

### 4.1.4. Moisture Adhesion Stability

Figure 7 shows the test results for Moisture Adhesion Stability. As a result of the Moisture Adhesion Stability test, materials G, H, and L were identified as having 10 points because there was no abnormality in the test piece under all conditions, and material I was identified as having 8 points due to floating in freezing melting and cold and hot repeatedly. In addition, material J was identified as having 7 points due to the occurrence of excitation in wet, frozen melting, and cold and hot repetition, and material K was identified as having 5 points due to excitation in all tests except for the dry surface.

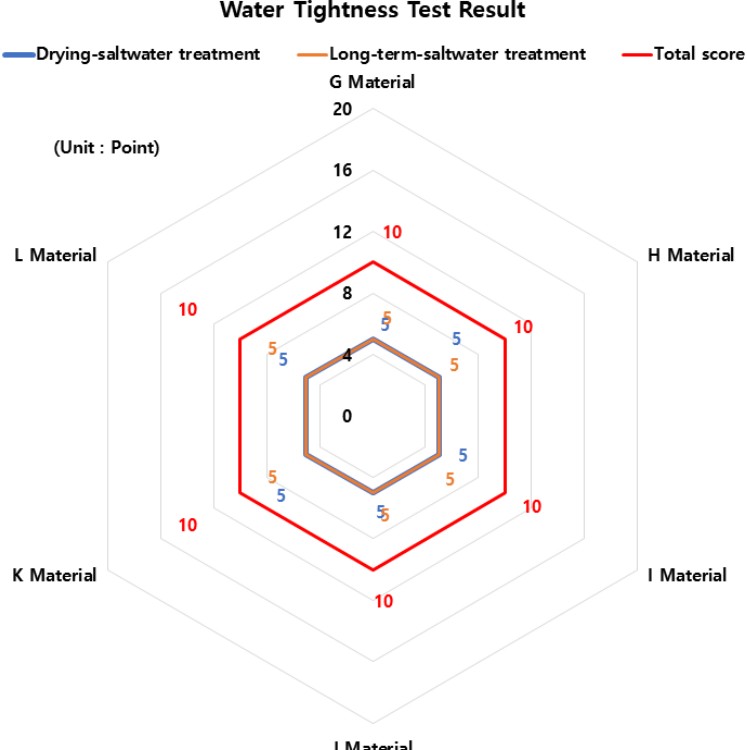

**Figure 7.** KCS 11 44 00: 2018 "Joint District" Moisture Adhesion Stability Test Results.

4.1.5. Water Tightness

The test results for water tightness are shown in Figures 8 and 9 below. As a result of the water tightness test, all materials were confirmed to have 10 points because no pitcher was generated in the test piece under all conditions.

**Figure 8.** KCS 11 44 00: 2018 "Joint District" Water Tightness Test Result.

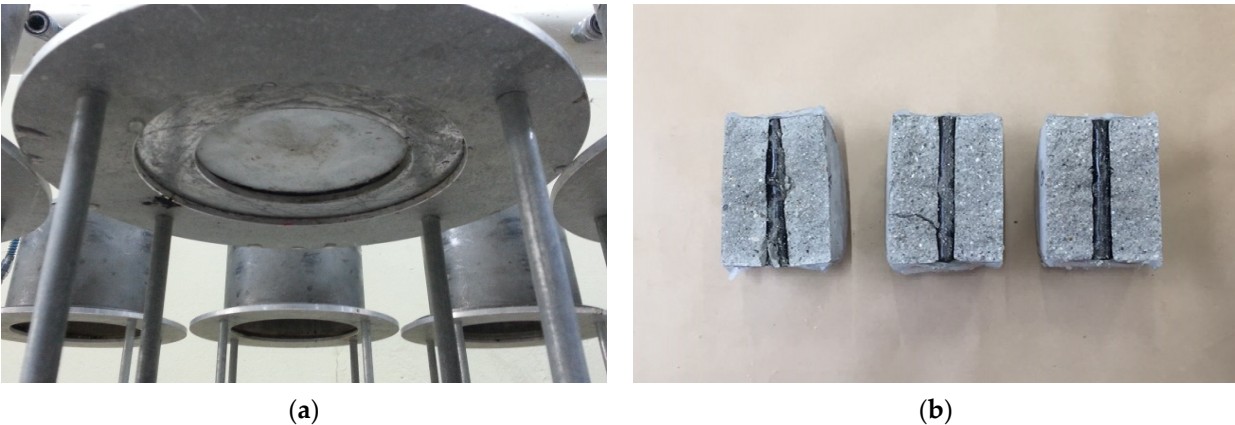

(**a**)　　　　　　　　　　　　　　　　(**b**)

**Figure 9.** KCS 11 44 00: 2018 "Joint District" Water Tightness Test Results. They should be listed as: (**a**) water tightness test; (**b**) water tightness test result.

4.1.6. Temperature Dependence

The test results for temperature dependence are shown in Figure 10. As a result of the temperature dependence test, all materials were confirmed to have 10 points because no permeation occurred in the test piece under all conditions.

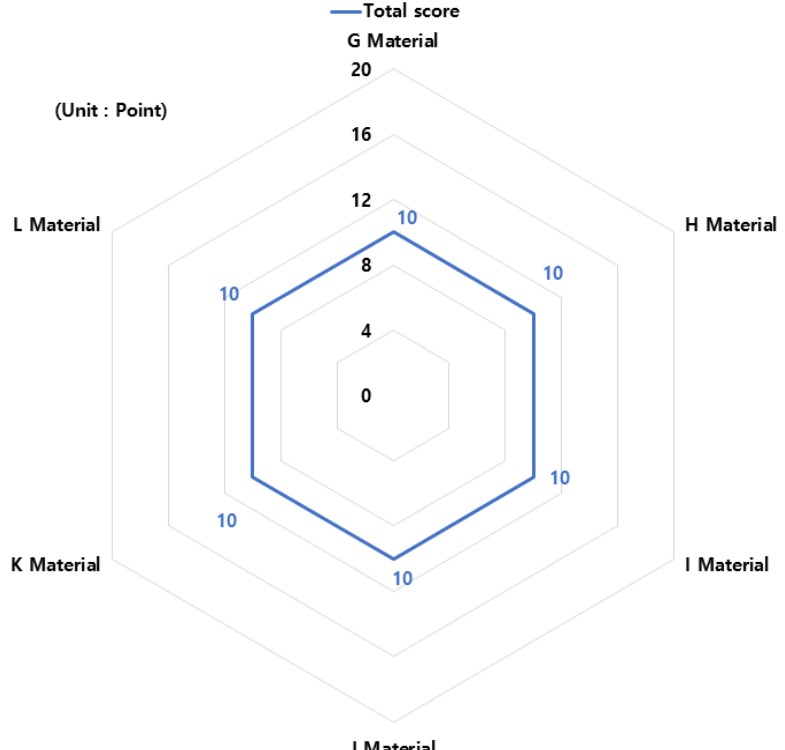

**Figure 10.** KCS 11 44 00: 2018 "Joint District" Temperature Dependency Test Results.

4.1.7. Crack Resistance

The test results for crack resistance are shown in Figures 11 and 12. As a result of the crack resistance test, all materials were confirmed to have 10 points because cracks and fractures did not occur in the test piece under all conditions.

**Crack Resistance Test Results**

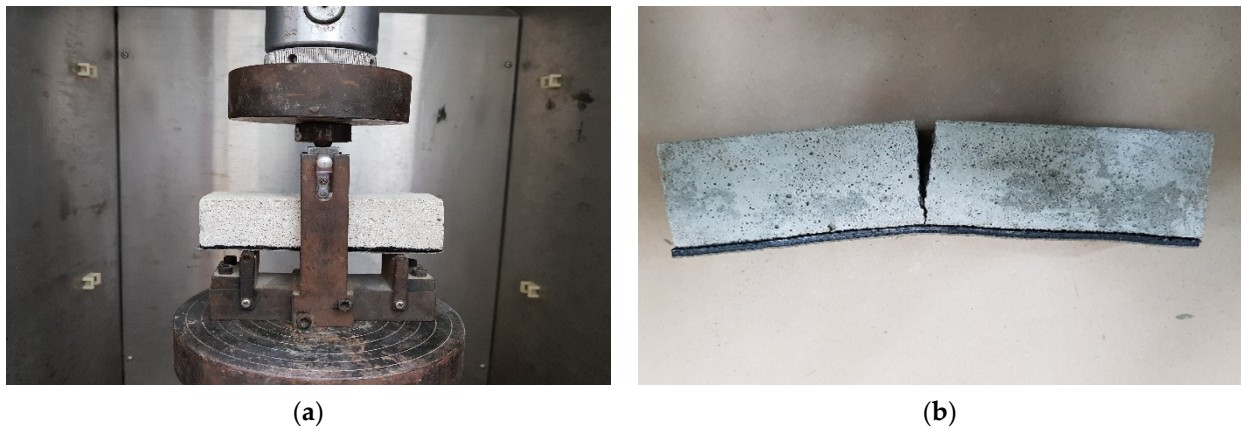

**Figure 11.** KCS 11 44 00: 2018 "Joint District" Crack Resistance Test Results.

|              |              |
|:------------:|:------------:|
| (**a**)      | (**b**)      |

**Figure 12.** KCS 11 44 00: 2018 "Joint District" Crack Resistance Test Results. They should be listed as: (**a**) Crack Resistance test; (**b**) Crack Resistance test result.

4.1.8. Durability

The test results for durability are shown in Table 5 and Figure 13. As a result of the durability test, all materials were found to have 10 points because they satisfied all of the quality standard items.

**Table 5.** KCS 11 44 00: 2018 "Joint District" Durability Test Results.

| Item | Test Results | Note |
|:----:|:-------------|:----:|
| Material G | (1) Korean Industrial Standard KS F 4917-16 "Modified Asphalt Waterproofing Sheet": satisfied all quality criteria<br>(2) Korean Industrial Standard KS F 4935-18 "Adhesive Flexible Rubber Asphalt-based Leakage Repair Injecting Sealant: satisfied all quality criteria | - |
|  | 10 Points |  |

**Table 5.** *Cont.*

| Item | Test Results | Note |
|---|---|---|
| Material H | (1) Korean Industrial Standard KS F 4917-16 "Modified Asphalt Waterproofing Sheet": satisfied all quality criteria<br>(2) Korean Industrial Standard KS F 3211-15 "Construction Waterproofing Membrane": satisfied all quality criteria | - |
| | 10 Points | |
| Material I | (1) Korean Industrial Standard KS F 4934-18 "Self-adhesive Rubberized Asphalt Waterproofing Sheet": satisfied all quality criteria | - |
| | 10 Points | |
| Material J | (1) Korean Industrial Standard KS F 4917-16 "Modified Asphalt Waterproofing Sheet": satisfied all quality criteria | - |
| | 10 Points | |
| Material K | (1) Korean Industrial Standard KS F 4911-19 "Synthetic Polymer Waterproofing Sheet": satisfied all quality criteria | - |
| | 10 Points | |
| Material L | (1) Korean Industrial Standard KS F 3211-15 "Construction Waterproofing Membrane": satisfied all quality criteria | - |
| | 10 Points | |

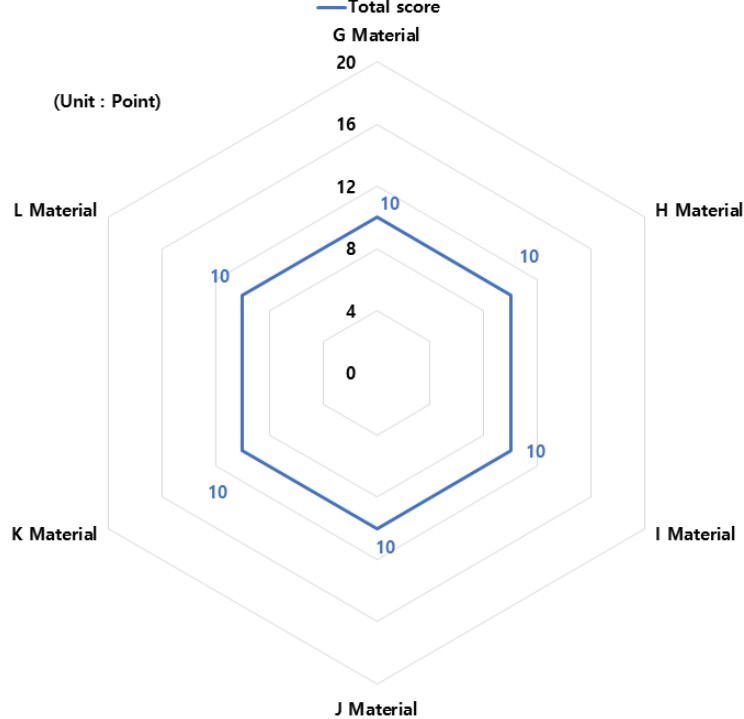

**Figure 13.** KCS 11 44 00: 2018 "Joint District" Durability Test Results.

4.1.9. Stability of Low-Temperature Adhesion

The test results for the stability of low-temperature adhesion are shown in Figures 14 and 15. As a result of the low-temperature adhesion stability test, all materials were confirmed to have 10 points because no pitcher was generated in the test piece under all conditions.

Low Temperature Adhesion Stability Test Results

— Total score

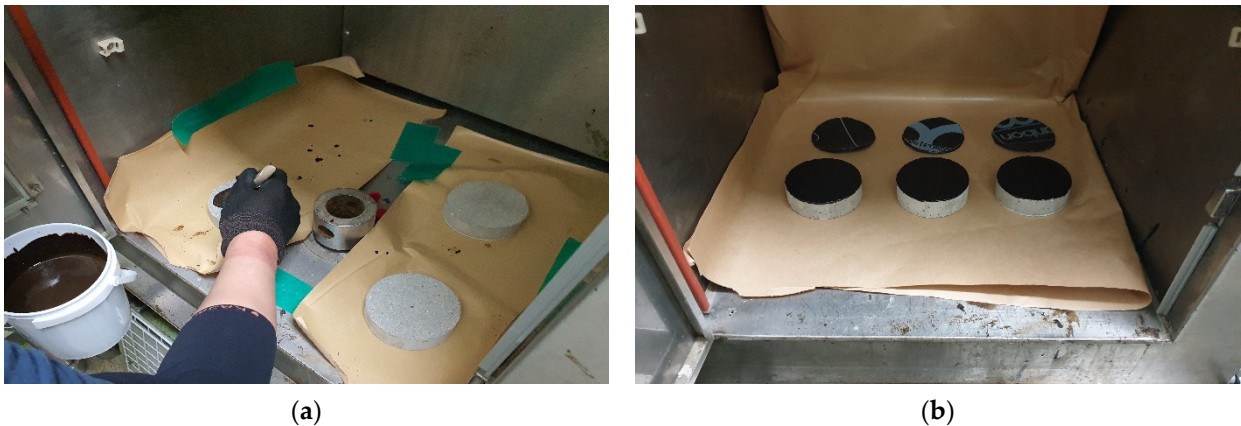

**Figure 14.** KCS 11 44 00: 2018 "Joint District" Low-Temperature Adhesion Stability Test Results.

(**a**)                                                      (**b**)

**Figure 15.** KCS 11 44 00: 2018 "Joint District" Low-Temperature Adhesion Stability Test Results. They should be listed as: (**a**) low-temperature bonding installation; (**b**) Low-Temperature Adhesion Stability Test Results.

4.1.10. KCS 11 44 00: 2018 "Joint District" Performance Evaluation Comprehensive

The comprehensive results of the KCS 11 44 00: 2018 "Joint District" performance evaluation are shown in Table 6, and Figures 16 and 17 below. According to the overall results of the performance evaluation, material G (composite waterproofing method (adhesive flexible seal material + improved asphalt waterproofing sheet)) was given 94 points out of 100 points and material H (composite waterproofing method (urethane coating waterproofing + improved asphalt sheet waterproofing)) was given 91 points. Material I (self-adhesive waterproof sheet (rubber asphalt or butyl rubber)) was given 89 points, and material J (improved asphalt water-proof sheet (rubber asphalt)) and material L (coating waterproof material (rubber asphalt or urethane rubber)) were given 76 points. Lastly, material E (synthetic polymer waterproof sheet (vinyl chloride resin)) was confirmed to have a score of 71.

**Table 6.** KCS 11 44 00: 2018 "Joint District" Overall Performance Evaluation Results.

| Number | Item | Material | | | | | | |
|---|---|---|---|---|---|---|---|---|
| | | Score | G | H | I | J | K | L |
| 1 | Chemical Resistance | 20 Points | 17 Points | 14 Points | 14 Points | 8 Points | 11 Points | 14 Points |
| 2 | Structural Response and Adaptability | 20 Points | 17 Points | 17 Points | 17 Points | 11 Points | 5 Points | 2 Points |
| 3 | Moisture Adhesion Stability | 10 Points | 10 Points | 10 Points | 8 Points | 7 Points | 5 Points | 10 Points |
| 4 | Water Tightness | 10 Points | 10 Points | 10 Points | 10 Points | 10 Points | 10 Points | 10 Points |
| 5 | Temperature Dependency | 10 Points | 10 Points | 10 Points | 10 Points | 10 Points | 10 Points | 10 Points |
| 6 | Crack Resistance | 10 Points | 10 Points | 10 Points | 10 Points | 10 Points | 10 Points | 10 Points |
| 7 | Durability | 10 Points | 10 Points | 10 Points | 10 Points | 10 Points | 10 Points | 10 Points |
| 8 | Low-Temperature Adhesion Stability | 10 Points | 10 Points | 10 Points | 10 Points | 10 Points | 10 Points | 10 Points |
| | Total Score | 100 Points | 94 Points | 91 Points | 89 Points | 76 Points | 71 Points | 76 Points |

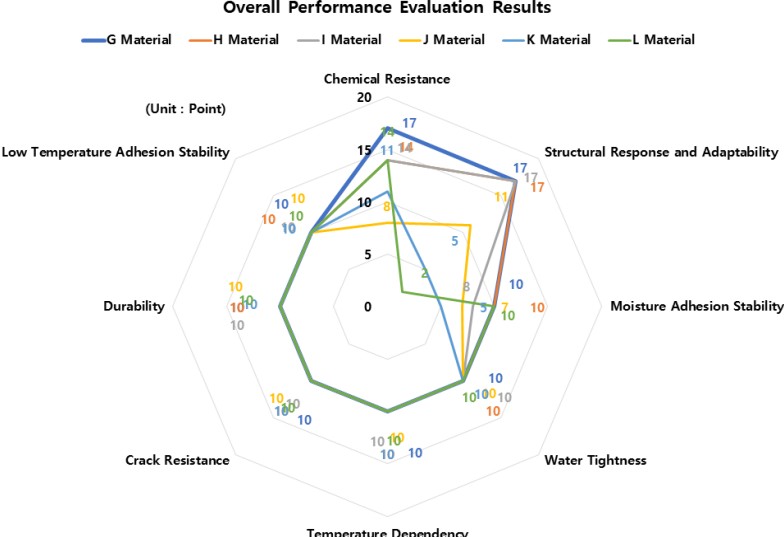

**Figure 16.** KCS 11 44 00: 2018 "Joint District" Overall Performance Evaluation Results.

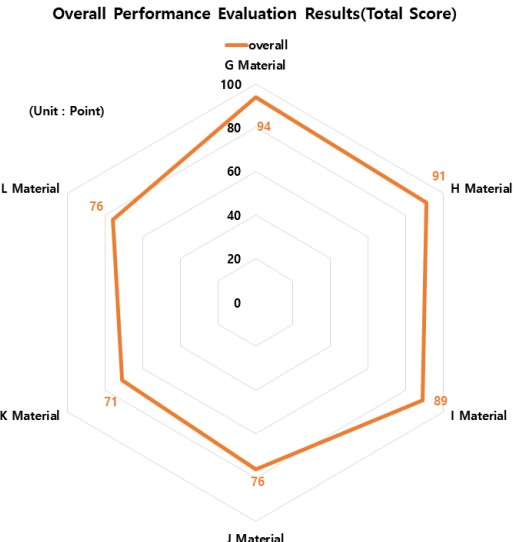

**Figure 17.** KCS 11 44 00: 2018 "Joint District" Overall Performance Evaluation Results (Total Score).

*4.2. Korea Industrial Standard KS F 4938-18 Test Result of "Root Penetration Resistance Test Method"*

4.2.1. Test Specimen Composition

The barometric performance test was conducted with the composition and test specimen using the following method as shown in Table 7.

**Table 7.** Diagram and Test Specimen for Root Penetration Resistance Performance by Waterproofing Methods.

| Item | Diagram and Test Specimen | | Item | Diagram and Test Specimen | |
|---|---|---|---|---|---|
| Material A | Diagram |  | Material G | Diagram |  |
| | Test Specimen |  | | Test Specimen |  |
| Material B | Diagram |  | Material H | Diagram |  |
| | Test Specimen |  | | Test Specimen |  |
| Material C | Diagram |  | Material I | Diagram |  |
| | Test Specimen |  | | Test Specimen |  |
| Material D | Diagram |  | Material J | Diagram |  |
| | Test Specimen |  | | Test Specimen |  |
| Material E | Diagram |  | Material K | Diagram |  |
| | Test Specimen |  | | Test Specimen |  |
| Material F | Diagram |  | Material L | Diagram |  |
| | Test Specimen |  | | Test Specimen |  |

### 4.2.2. Results of Root Penetration Resistance Performance Test

Table 8 shows the results of the root penetration resistance performance test. As a result of two years of observation, materials A to F installed with the root layer did not find any root penetration or change in the waterproof layer, and materials G to L without the root layer protruded from the general part or penetrated the joint and general part.

**Table 8.** Root Penetration Resistance Performance Test results.

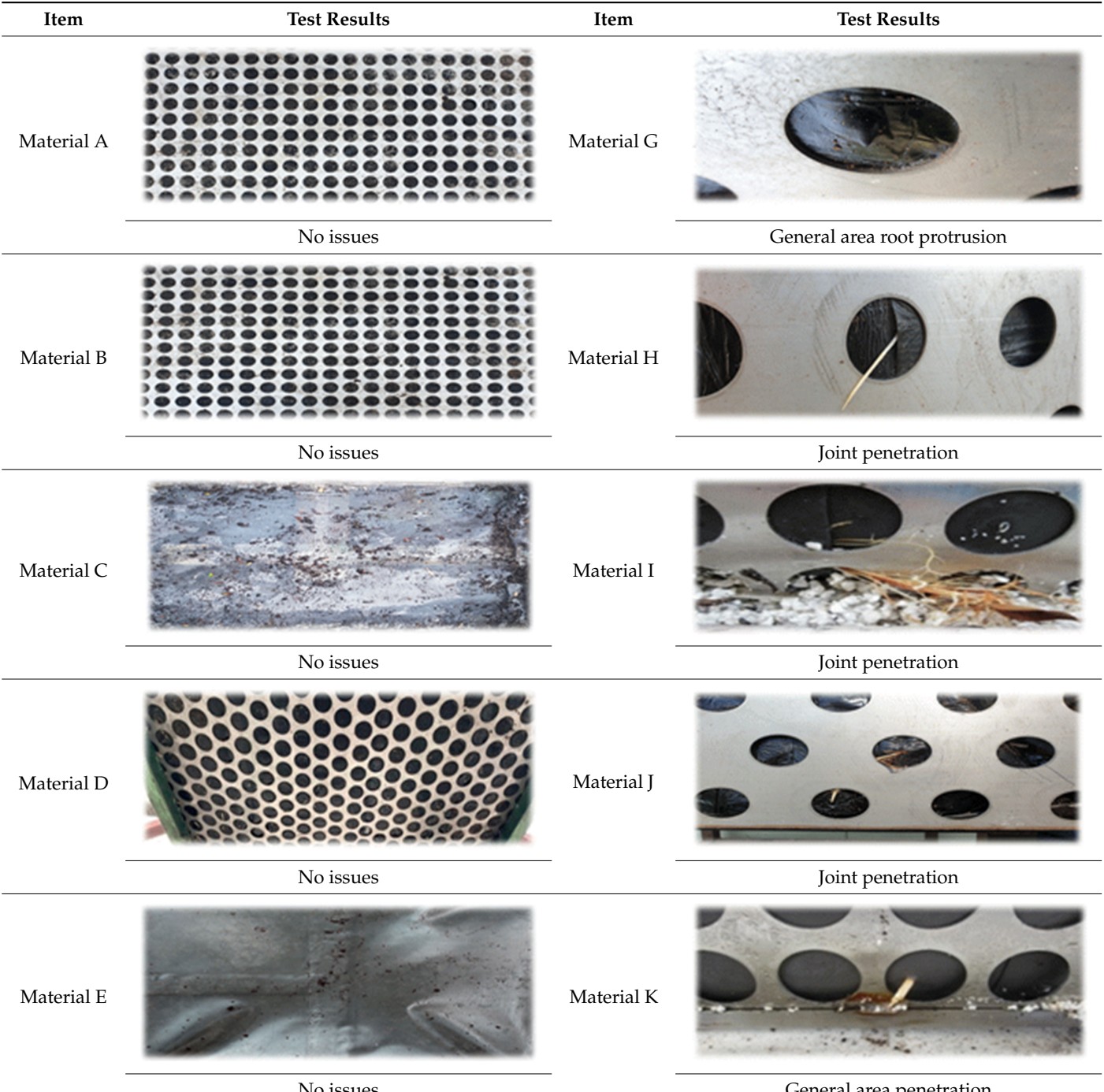

| Item | Test Results | Item | Test Results |
|------|-------------|------|-------------|
| Material A | No issues | Material G | General area root protrusion |
| Material B | No issues | Material H | Joint penetration |
| Material C | No issues | Material I | Joint penetration |
| Material D | No issues | Material J | Joint penetration |
| Material E | No issues | Material K | General area penetration |

**Table 8.** *Cont.*

| Item | Test Results | Item | Test Results |
|------|--------------|------|--------------|
| Material F | 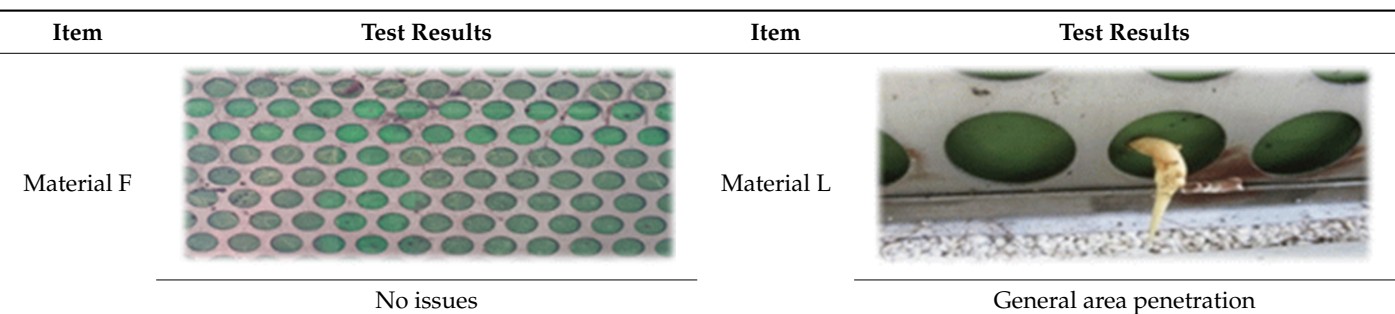 | Material L | |
| | No issues | | General area penetration |

### 4.3. Consideration

Based on the above performance evaluation results, the score deviations of material G, material H, and material I were not large and similar, and the score deviation of material J, material K, and material L were found to be similar. In addition, it was confirmed that defects in the waterproof layer occurred depending on the presence or absence of the waterproof layer regardless of the material group. Accordingly, the rating of the upper slab on the top floor of the apartment building needs to be calculated by classifying waterproof methods without a barrier layer into grades 3 and 4, and waterproof methods with a barrier layer into grades 1 and 2. The results of the grading setting accordingly are shown in Table 9.

**Table 9.** Overall Performance Rating Determination Results.

| Item | Method Configuration | | Performance Evaluation Score | Ranking | Grading | Note |
|------|---------|-----------------------------------------------|------------|---------|---------|------|
| | **Method** | **Presence of Root Penetration Resistance Layer** | | | | |
| Material A (Composite Waterproofing Material (Adhesive Flexible Sealant + Modified Asphalt Sheet Waterproofing) + Root Penetration Resistance Layer) | Composite Waterproofing | | 94 | 1 | | - |
| Material B (Composite Waterproofing Method (Polyurethane Membrane Waterproofing + Modified Asphalt Sheet Waterproofing) + Root Penetration Resistance Layer) | Composite Waterproofing | | 91 | 2 | ★★★★ (Grade 1) | - |
| Material C (Self-adhesive Waterproofing Sheet (Rubber Asphalt-based or Butyl Rubber-based) + Root Penetration Resistance Layer) | Sheet Waterproofing | Yes | 89 | 3 | | - |
| Material D (Modified Asphalt Waterproofing Sheet (Rubber Asphalt-based) + Root Penetration Resistance Layer) | Sheet Waterproofing | | 76 | 4 | | - |
| Material E (Synthetic Polymer Waterproofing Sheet (Vinyl Chloride Resin-based) + Root Penetration Resistance Layer) | Sheet Waterproofing | | 71 | 5 | ★★★ (Grade 2) | - |
| Material F (Membrane Waterproofing Material (Rubber Asphalt-based or Polyurethane Rubber-based) + Root Penetration Resistance Layer) | Sheet Waterproofing | | 76 | 4 | | - |

**Table 9.** *Cont.*

| Item | Method Configuration | | Performance Evaluation Score | Ranking | Grading | Note |
|---|---|---|---|---|---|---|
| | Method | Presence of Root Penetration Resistance Layer | | | | |
| Material G (Composite Waterproofing Material (Adhesive Flexible Sealant + Modified Asphalt Sheet Waterproofing)) | Composite Waterproofing | | 94 | 6 | | - |
| Material H (Composite Waterproofing Method (Polyurethane Membrane Waterproofing + Modified Asphalt Sheet Waterproofing)) | Composite Waterproofing | | 91 | 7 | ★★ (Grade 3) | - |
| Material I (Self-adhesive Waterproofing Sheet (Rubber Asphalt-based or Butyl Rubber-based)) | Sheet Waterproofing | No | 89 | 8 | | - |
| Material J (Modified Asphalt Waterproofing Sheet (Rubber Asphalt-based)) | Sheet Waterproofing | | 76 | 9 | | - |
| Material K (Synthetic Polymer Waterproofing Sheet (Vinyl Chloride Resin-based)) | Sheet Waterproofing | | 71 | 10 | ★ (Grade 4) | - |
| Material L (Membrane Waterproofing Material (Rubber Asphalt-based or Polyurethane Rubber-based)) | Sheet Waterproofing | | 76 | 9 | | - |

## 5. Conclusions

The conclusions of this study are as follows:

1. According to the performance evaluation of the apartment building, the waterproof method applied to the upper slab on the top floor of the underground structure of the apartment building was excellent, and some sheet waterproof methods were found to have similar performance to or lower performance than the coating waterproof method. Self-adhesive waterproof sheets with excellent adhesion performance are judged to be highly responsive to the underground environment, and synthetic polymer waterproof sheets constructed using adhesives have lower scores than coating waterproofing methods due to problems such as adhesion due to the underground wet environment.

2. The performance score of the same asphalt-based self-adhesive waterproof sheet and the improved asphalt waterproof sheet differed in the performance score in the joint performance evaluation, due to the deterioration of the physical properties and construction quality of the sheet.

3. Accordingly, it is necessary to grade the waterproof performance by classifying the improved asphalt waterproof sheet, synthetic polymer waterproof sheet, and coating waterproof material into the same performance grade.

4. In the case of the composite waterproof method without a barrier layer, it received excellent performance scores in the performance evaluation of the apartment building but, if a separate barrier layer is not constructed, it still acts as a waterproof defect. Therefore, in order to prevent defects due to this cause in advance, it is reasonable to mark them at a lower performance grade than the performance grade with a barrier layer.

This study focused on developing a waterproof performance rating method specifically for the upper slab of the top floor in the underground structure of apartments, aligning

with the performance rating approach used for apartment houses in Korea. The findings are anticipated to serve as fundamental data for a universal waterproof performance rating applicable to various concrete building structures, taking into account evaluation methods and standards outlined by International Standards (ISO).

**Author Contributions:** Conceptualization, Methodology, Laboratory work, Data curation, Writing—original draft preparation, K.-w.A.; Conceptualization, Methodology, Writing—reviewing and editing, B.-i.K. All authors have read and agreed to the published version of the manuscript.

**Funding:** This research received no external funding.

**Data Availability Statement:** The data presented in this study are available on request from the corresponding author.

**Acknowledgments:** This study was financially supported by the Seoul National University of Science & Technology.

**Conflicts of Interest:** The authors declare no conflict of interest for the publication of the review paper.

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
