# Peer review of "A Study on Verification of Waterproofing Method Properties for Performance Grading in Apartment Houses-Upper Slab of the Underground Structure"

_buildings, doi:10.3390/buildings13092164_

Round 1

Reviewer 1 Report

The article addresses a topic of great relevance, however, the structure and organization need to be revised.

The methodology and results need to be highlighted. Furthermore, the manuscript contains an excessive number of tables, which impairs the fluency of reading and understanding of the content. This needs to be adjusted.

Author Response

The author revised the whole paper about the comments. The author would appreciate it if you could check the attached document and revised paper.

Reviewer 2 Report

This manuscript could be accepted after a minor revision. Main concerns are listed as follows:

1.      In the Fig 3-7 of this manuscript, only a few samples after experiment are presented without any explanations. It is very confusable to readers since we have no idea on which kind of material being presented. And it is nor comprehensive to present only a limit amount of samples after experiment.

2.      How many types of materials are experimented and analyzed in this manuscript? Why 12 materials are presented in section 1 and section 4.2.1 while data of 6 materials are presented in other sections?

3.      Too many tables with many description are presented in this manuscript, it is very low efficient for readers to understand what you present. It might be better for authors to shorten those tables with key information being outlined, for example numbers.

4.     The introduction part is not comprehensive. I suggest that, the author extend Introduction with more discussions: A coupled thermal-elastic-plastic-damage model for concrete subjected to dynamic loading. International Journal of Plasticity, Implementation of Johnson-Holmquist-Beissel model in four-dimensional lattice spring model and its application in projectile penetration. International Journal of Impact Engineering, A bounding surface plasticity model for expanded polystyrene-sand mixture, Transportation Geotechnics

This manuscript could be accepted after a minor revision. Main concerns are listed as follows:

1.      In the Fig 3-7 of this manuscript, only a few samples after experiment are presented without any explanations. It is very confusable to readers since we have no idea on which kind of material being presented. And it is nor comprehensive to present only a limit amount of samples after experiment.

2.      How many types of materials are experimented and analyzed in this manuscript? Why 12 materials are presented in section 1 and section 4.2.1 while data of 6 materials are presented in other sections?

3.      Too many tables with many description are presented in this manuscript, it is very low efficient for readers to understand what you present. It might be better for authors to shorten those tables with key information being outlined, for example numbers.

4.     The introduction part is not comprehensive. I suggest that, the author extend Introduction with more discussions: A coupled thermal-elastic-plastic-damage model for concrete subjected to dynamic loading. International Journal of Plasticity, Implementation of Johnson-Holmquist-Beissel model in four-dimensional lattice spring model and its application in projectile penetration. International Journal of Impact Engineering, A bounding surface plasticity model for expanded polystyrene-sand mixture, Transportation Geotechnics

Author Response

(The authors gave the same response as above.)

Reviewer 3 Report

Do all changes highlighted in the attached paper .

The whole paper needs proofreading and some sentences need to be re-written correctly

Author Response

(The authors gave the same response as above.)

Reviewer 4 Report

1. In the introduction, what are the differences and limitations of guidelines and post-treatment measures for leakage defects in apartment buildings used for underground space leakage?

2. The paper shows a lack of innovation, with the research relying on established standards and evaluation methods that are devoid of theory. Furthermore, the final grade classification is excessively subjective and problematic in terms of scientific validity.

3. There are numerous spelling mistakes, such as “be-low” and “meth-od” in subsection 3.1 and “Subsubsection” in subsection 3.2.1. The wordings used are repeated, such as “by applying and applying”, it is recommended to check the whole text and revise it.

4. Unharmonized formatting of fonts in the tables, especially unit symbols, such as in table 3.

5. The use of quotation marks, brackets, braces, parenthesises and other symbols is a matter of formatting standardization, such as “(composite waterproofing method (adhesive flexible seal material + improved asphalt waterproofing sheet)”, “[synthetic polymer water-proof sheet (vinyl chloride resin)]”, etc. Recommended to check the whole text.

6. Outstanding grammatical problems, such as “E material [synthetic polymer water-proof sheet (vinyl chloride resin)] It was confirmed to be 71 points.”

7. Only the scoring results of the various materials are presented, the analysis of the test results is shallow, and the profundity of overall research is insufficient.

8. Redundant sentences, such as in the conclusion “This study studied the waterproof performance rating method only for the upper slab on the top floor of the underground structure of the apartment according to the performance rating method of the apartment house in Korea.” Suggested to condense every paragraph.

9. The reference format is not consistent.

Language expression lacks standardization and conciseness, there are several grammatical mistakes, and the content could be further condensed and improved. The specific modification suggestions can be found in “Comments and Suggestions”.

Author Response

(The authors gave the same response as above.)

Round 2

Reviewer 1 Report

The authors responded to my comments and the paper can now be accepted.

Reviewer 3 Report

Thank you for making all changes highlighted in the previously attached paper

Thank

Reviewer 4 Report

Revised article is well formatted and logical with no revision comments.